# Metabolic Profiling, Tissue Distribution, and Tolerance Assessment of Bopu Powder in Laying Hens Following Long-Term Dietary Administration

**DOI:** 10.3390/vetsci12090848

**Published:** 2025-09-02

**Authors:** Hongting Wang, Xinhao Wang, Jiaxin Xu, Zihui Yang, Zhen Dong, Jianguo Zeng, Hua Liu

**Affiliations:** 1HunanProvince Key Laboratory of Traditional Chinese Veterinary Medicine, College of Veterinary Medicine, Hunan Agricultural University, Changsha 410128, China; 322860@stu.hunau.edu.cn (H.W.); wxh995651051@stu.hunau.edu.cn (X.W.); 15588521035@163.com (J.X.); yangzihui_2021@hunau.edu.cn (Z.Y.); dongzhen2022@stu.hunau.edu.cn (Z.D.); 2Chinese Medicinal Materials Breeding Innovation Centre of Yuelushan Laboratory, Changsha 410128, China

**Keywords:** Bopu Powder, laying hens, metabolism, residues, safety evaluation

## Abstract

This study assessed the safety and metabolism of Bopu Powder—a herbal veterinary product—in laying hens following 56 days of dietary supplementation. Its major alkaloids, protopine and allocryptopine, were extensively metabolized. Residual levels were low and limited to the liver and kidneys, with occasional detection in eggs only at the highest dose (500 mg/kg). No residues were found in muscle, fat, or other edible tissues. Serum biochemistry and histopathology revealed no adverse effects, supporting the safety of Bopu Powder for long-term use in laying hens.

## 1. Introduction

The poppy family plant *Macleaya cordata* (Willd.) R. Br. is rich in alkaloid compounds, with sanguinarine (SAN), chelerythrine (CHE), allocryptopine (ALL), and protopine (PRO) being the most abundant [1]. Modern pharmacological studies have demonstrated that Macleaya cordata extracts exhibit diverse biological activities, including antibacterial [2], anti-inflammatory [3], and antitumor effects [4,5]. Bopu Powder, a new veterinary drug formulation primarily composed of PRO and ALL as its main active ingredients, has been approved for treating *Escherichia coli*-induced diarrhea in chickens [6]. Furthermore, evidence suggests that dietary supplementation with Bopu Powder can enhance growth performance in broilers [7] and improve egg quality in laying hens [8]. However, evidence indicates that PRO/ALL exhibit dose-dependent toxicity, with acute exposure causing pulmonary hemorrhage and hepatorenal injury in mice [9,10], although the mechanisms underlying their chronic toxicity remain unclear. Given the potential cardiotoxicity of PRO, phytopharmaceutical preparations containing PRO/ALL may pose potential health risks to both humans and animals [11].

Guo et al. [12] investigated the tissue distribution of PRO in *Dactylicapnos scandens* using LC-ESI-MS/MS. Following a single oral administration of D. scandens extract (126.6 mg/kg) to rats, PRO residues were highest in the small intestine (9.31 ± 2.45 μg/g), followed by the stomach (2.91 ± 0.19 μg/g) and kidneys (2.33 ± 0.71 μg/g). PRO concentrations fell below the limit of quantification (LOQ, 10 ng/mL) within 8 h post-administration, indicating rapid systemic elimination. Huang et al. [13] employed HPLC-QTOF-MS to study the residues of PRO and ALL in rats. After PRO administration, the highest residual levels were observed in the spleen at 24 h (44.04 ng/g) and the liver at 48 h (20.64 ng/g). Following ALL administration, residues were detected in urine, feces, plasma, and various tissues, albeit at consistently low levels (< 3 ng/g). The highest residue level after PRO administration was found in the liver at 48 h (61.8 ng/g), with levels in other tissues below 6.5 ng/g. Liu et al. [14] examined the residual distribution of PRO and ALL in laying hens’ tissues following 7 days of dietary supplementation with Bopu Powder. At a dose of 400 mg/kg feed, PRO residues in eggs peaked on day 7 (2.8 μg/kg) but became undetectable 5 days post-withdrawal; ALL remained undetectable throughout. At the higher dose of 2000 mg/kg feed, residual levels of PRO and ALL reached 16.72 μg/kg and 8.61 μg/kg, respectively, 1 day after withdrawal. PRO was undetectable 7 days post-withdrawal, while ALL residues persisted. Tissue analysis revealed significantly elevated concentrations of both compounds 2 h after withdrawal, followed by a rapid decline within 24 h. No residues were detected in any tissues by 7 days post-withdrawal. The study indicated a low residue risk associated with the 400 mg/kg dose. However, the 2000 mg/kg dose may elevate the residue risk, necessitating a withdrawal period of at least 7 days to ensure food safety.

Currently, research on residues of Bopu Powder in laying hens has focused solely on short-term exposure (7 days) for therapeutic purposes. Short-term exposure studies, while valid for establishing therapeutic withdrawal periods, are critically limited for two reasons: Firstly, they fail to account for the potential of low-level residue accumulation in metabolic tissues (e.g., liver, kidneys) during continuous intake. Secondly, they cannot assess the steady-state residue deposition in eggs or the potential for delayed physiological effects under long-term, zootechnical use conditions. Currently, there remains a lack of systematic evaluation regarding the safety of long-term dietary administration of Bopu Powder throughout the entire laying cycle of hens. To address this data gap, we hypothesize that even prolonged exposure at doses as high as 500 mg/kg will not lead to toxicologically significant residue accumulation of its primary active constituents, PRO and ALL, in edible tissues and eggs, nor will it cause significant adverse physiological effects in laying hens. To test this hypothesis, this study systematically characterizes, for the first time, the metabolic profiles of PRO and ALL in laying hens; quantitatively assesses their residual distribution across multiple tissues; and comprehensively evaluates their safety and tolerability following 56 days of continuous administration through integrated serum biochemical parameters and histopathological examinations. Ultimately, this research aims to elucidate the transformation and residue patterns of PRO and ALL from Bopu Powder in laying hens, thereby providing critical scientific evidence to support the safety of its long-term use.

## 2. Materials and Methods

### 2.1. Chemical Reagent

Bopu Powder (Production Batch No.: 190701; specification: total PRO and ALL content not less than 1.5%, with PRO content not less than 1.0%; supplied by Hunan MICOLTA Biological Resources Co., Ltd., Changsha, China.). PRO (Batch No.: 140701, purity ≥ 98.0%; Hunan MICOLTA Biological Resources Co., Ltd., Changsha, China.). ALL (Batch No.: 170601, purity ≥ 98.0%; Hunan MICOLTA Biological Resources Co., Ltd., Changsha, China.). Sodium barbiturate (purity ≥ 97.0%, Merck KGaA, Darmstadt, Germany). Dimethyl sulfoxide (DMSO; Tedia, Fairfield, OH, USA). Acetonitrile and methanol (HPLC grade; Merck KgaA, Germany). Formic acid (LC-MS grade; Tedia, Fairfield, OH, USA). Tetrahydropalmatine standard (TET, Batch No.: 110726-201819, purity 99.8%; National Institutes for Food and Drug Control, NIFDC, Beijing, China). Basal diet (Hunan Jinpai Haotian Biotechnology Co., Ltd., Changsha, China). Physiological saline (Anhui Shuanghe Pharmaceutical Co., Ltd., Wuhu, China). Ultrapure water was prepared in-house using a Milli-Q system (MilliporeSigma, Burlington, MA, USA).

### 2.2. Solution Preparation

Saline Solution (0.75%): A 0.75% saline solution was prepared by diluting 250 mL of 0.9% saline solution with 50 mL of sterile ultrapure water to a final volume of 300 mL. Dosing Solution: PRO and ALL monomers (2.00 mg each) were accurately weighed and dissolved in 200 μL of dimethyl sulfoxide (DMSO) via sonication. Subsequently, 2 mL of the 0.75% saline solution was added to dilute the mixture. Control Solution: The control solution was prepared by combining 2 mL of the 0.75% saline solution with 200 μL of DMSO, followed by thorough mixing.

### 2.3. Birds and Grouping

A total of 333 sixty-five-week-old Roman Gray laying hens, exhibiting a laying rate of 85.75 ± 8.35%, were obtained from the Hunan Institute of Animal Science and Veterinary Medicine. Throughout the experimental period, the hens were housed in cages under standard laboratory conditions, with ad libitum access to feed and water. Following a 7-day acclimatization period, standardized feeding regimens and routine health management procedures, including biosecurity protocols, were implemented. The nutritional composition of the diet was formulated in accordance with the Chinese National Standard GB/T 5916-2020 [15] (“Formula Feed for Laying Hens and Broilers”) and contained no added medications; its detailed nutritional profile is presented in Table 1. Nine hens were allocated to the metabolic study. These birds were randomly divided into three groups (*n* = 3 per group): the blank control group, the PRO group, and the ALL group. Test compounds were administered via intravenous injection at a dose of 1 mg/kg body weight (bw).The remaining 324 hens were utilized for tolerance and residue studies, conducted in compliance with the “Guidelines for Target Animal Tolerance Testing of Feed and Feed Additives in Livestock and Poultry” issued by the Ministry of Agriculture and Rural Affairs [16], P.R. China. Three experimental groups were established: CN (control, no additive), BP (50 mg/kg additive), and BPX (500 mg/kg additive). The study employed a randomized design with 6 replicates per group and 18 hens per replicate. The feeding trial duration was 56 days. This study received approval from the Animal Ethics Committee of Hunan Agricultural University (Approval No.: HUNAULSK 20230807–03).

### 2.4. Sample Collection

#### 2.4.1. Identification of Metabolites for Allocryptopine and Protopine in Laying Hens

Plasma: Blood samples were collected from the subcutaneous ulnar vein at 5, 10, 20, 30, 60, and 120 min post-administration. Samples were centrifuged at 4 °C and 3200× *g* for 10 min. The supernatant (plasma) was collected and stored at −80 °C for subsequent metabolite analysis. Additionally, at the end of week 8, from each replicate group, one hen was randomly selected. A 10 mL blood sample was collected from the subcutaneous ulnar vein; 5 mL was transferred into an EDTA-K2 anticoagulant tube, gently mixed, and centrifuged at 3000× *g* for 10 min. The separated plasma was transferred into EP tubes (Eppendorf tubes) and stored at −20 °C for residue analysis. The remaining 5 mL was transferred into a plain serum tube (without additives). Following a 30 min clotting period at room temperature (with the tube inclined), the sample was centrifuged at 3000× *g* for 10 min. The supernatant (serum) was collected into EP tubes and stored at −20 °C for serum biochemical parameter analysis.

Feces: Feces were collected over the intervals 0–6 h, 6–12 h, 12–24 h, and 24–48 h following intravenous administration. All fecal samples were immediately stored at −80 °C.

#### 2.4.2. Quantification of Protopine and Allocryptopine in Laying Hens’ Tissues and Eggs

Eggs: During weeks 4 and 8 of the trial, two eggs per replicate group, approximating the mean egg weight, were collected and stored at 4 °C until analysis.

Tissues: At week 8 of the trial, one laying hen per replicate was euthanized via intravenous sodium barbiturate injection following blood collection, in strict adherence to the *AVMA Guidelines for the Euthanasia of Animals: 2020 Edition* [17]. Necropsy was performed to isolate cardiac, hepatic, splenic, renal, ovarian, oviductal, and intestinal tissues. Segments of jejunum, ileum, liver, ovary, oviduct, uterus, pectoral muscle, thigh muscle, gizzard, abdominal fat, and subcutaneous fat were individually wrapped in aluminum foil, flash-frozen in liquid nitrogen, and stored at −80 °C pending analysis. Concurrently, samples of hepatic, renal, oviductal, and uterine tissues were preserved in 10% neutral buffered formalin for histopathological processing.

#### 2.4.3. Serum Biochemistry

At the end of the 8th week, one laying hen was randomly selected from each replicate, and 5 mL of blood was collected from the wing vein using additive-free blood collection tubes. The tubes were inclined at an angle and allowed to clot for 30 min, followed by centrifugation at 3000 r/min for 10 min. The supernatant serum was carefully collected, transferred into EP tubes, and stored at −20 °C for subsequent serum biochemical analysis.

### 2.5. Sample Pretreatment

Plasma: Plasma samples stored at −80 °C were thawed at room temperature. A pooled plasma sample was prepared by combining equal volumes of plasma from each designated collection timepoint. A 200 μL aliquot of this pooled plasma was precisely transferred into a 1.5 mL microcentrifuge tube. Subsequently, 800 μL of acetonitrile was added to precipitate proteins, and the mixture was vortex-mixed vigorously for 2 min. Following protein precipitation, the tube was sealed and sonicated for 20 min. The sample was then centrifuged at 12,000× *g* for 10 min. The resulting supernatant was carefully collected and evaporated to dryness under a gentle stream of nitrogen at room temperature. The dried residue was reconstituted in 200 μL of acetonitrile, vortex-mixed for 2 min, and centrifuged again at 12,000× *g* for 5 min. Finally, the supernatant was filtered through a 0.22 μm organic membrane filter prior to LC-MS/MS analysis.

Feces: Fecal samples stored at −80 °C were thawed at room temperature and homogenized. A 2.00 g aliquot of the homogenized feces was accurately weighed into a 50 mL centrifuge tube. Ten milliliters of acetonitrile, serving as the extraction solvent, was added to the sample, which was then vortex-mixed for 5 min. The tube was sealed and subjected to sonication for 20 min. The mixture was subsequently centrifuged at 10,000× *g* and 4 °C for 8 min. The supernatant was collected and dried under a nitrogen stream at room temperature. The residue was reconstituted in 0.5 mL of acetonitrile, vortex-mixed for 1 min, and centrifuged at 12,000× *g* for 10 min. The obtained supernatant was filtered through a 0.22 μm organic membrane filter before instrumental analysis.

Eggs: Two grams of homogenized whole egg liquid was transferred into a 50 mL centrifuge tube containing homogenization beads. Twenty microliters of the 1 mg/L internal standard (IS) working solution was precisely added, followed by vortex mixing for 30 s. Subsequently, 10 mL of acetonitrile was added and the mixture was vortex-mixed for 5 min to facilitate extraction. The sample was then centrifuged at 8000× *g* for 5 min. Five milliliters of the resulting supernatant was transferred to a ten-milliliter centrifuge tube, combined with three milliliters of n-hexane, and vortex-mixed for 1 min. After phase separation, the upper n-hexane layer was carefully removed using a rubber-bulb pipette. Then, 1 mL of the lower aqueous acetonitrile phase was collected, evaporated to dryness under a nitrogen stream, and reconstituted in 0.5 mL of acetonitrile with 30 s of vortex mixing. The final extract was filtered through a 0.22 μm organic solvent-compatible membrane prior to LC-MS/MS analysis. Only two quality control (QC) measures were implemented: parallel duplicate samples, and blank/carryover checks.

Tissues: Tissue samples stored at −80 °C were thawed to room temperature. Approximately 2 g of tissue was weighed into a 50 mL screw-cap centrifuge tube. Twenty microliters of freshly prepared 20 mg·L^−1^ IS solution and 1 mL of physiological saline were added. The mixture was homogenized using a high-speed homogenizer for 30 s to form a tissue homogenate. Twenty milliliters of ice-cold extraction solvent consisting of 1% formic acid in acetonitrile–water (95:5, *v*/*v*) was added to precipitate proteins, followed by 1 min of vortex mixing. The tube opening was sealed with plastic wrap, and the sample was sonicated for 20 min. Centrifugation was performed at 4 °C and 3200× *g* for 10 min. Five milliliters of the supernatant was transferred to a ten-milliliter centrifuge tube, mixed with two milliliters of n-hexane, and vortexed for 30 s. After centrifugation at 4 °C and 3200× *g* for 5 min, the upper lipid layer was aspirated and discarded using a pipette. One milliliter of the supernatant was transferred to a clean ten-milliliter tube and evaporated to dryness under nitrogen at 35 °C. The residue was reconstituted in 1 mL of 1% formic acid in methanol, vortex-mixed for 30 s, and centrifuged at 4 °C and 3200× *g* for 10 min. The clarified solution was filtered through a 0.22 μm membrane before instrumental analysis. Only two QC measures were implemented: parallel duplicate samples, and blank/carryover checks.

### 2.6. LC-MS Conditions

#### 2.6.1. LC-QTOF-MS Conditions

Chromatographic separation was performed on a Waters BEH C18 column (2.1 mm × 100 mm, 1.7 μm particle size) using a mobile phase consisting of 0.1% formic acid in water (A) and acetonitrile (B). The column temperature was maintained at 35 °C, and the sample tray was set to 4 °C. The following gradient elution program was employed: 8% B (0–10 min), linearly increased to 20% B (10–15 min), ramped to 85% B (15–25 min), stepped to 95% B (25–25.1 min), held at 95% B (25.1–29.5 min), returned to 8% B (29.5–30 min), and re-equilibrated at 8% B (30–35 min). The injection volume was 5 μL.

Mass spectrometric detection was conducted in positive electrospray ionization (ESI+) mode with the following optimized parameters: capillary voltage, 4000 V; nebulizer gas pressure, 35 psi; drying gas temperature, 350 °C; drying gas flow rate, 12 L·min^−1^; sheath gas temperature, 350 °C; sheath gas flow rate, 11 L·min^−1^. High-purity nitrogen served as both the drying gas (≥99.9%) and the collision gas (≥99.999%).

#### 2.6.2. LC-QQQ-MS Conditions

Quantification of ALL and PRO in laying hens’ tissues and eggs was performed according to the method established by Liu et al. [14,18]. Chromatographic separation was achieved using an Agilent ZORBAX SB-C18 column (2.1 mm × 50 mm i.d., 1.8 μm particle size) maintained at 35 °C. The mobile phase consisted of 0.1% aqueous formic acid (A) and 0.1% formic acid in acetonitrile (B), delivered at a flow rate of 0.3 mL/min. The gradient elution program was executed as follows: 15% B (0–4 min), increased linearly to 17% B (4–5 min), ramped to 20% B (5–6 min), elevated to 35% B (6–12 min), stepped to 95% B (12–12.1 min), maintained at 95% B (12.1–14 min), returned to 15% B (14–14.1 min), and re-equilibrated at 15% B (14.1–17 min). A 1 μL aliquot was injected for analysis.

Mass spectrometric detection employed an electrospray ionization (ESI) source operating in positive ion mode under the following optimized parameters: capillary voltage, 4000 V; electron multiplier voltage (EMV), 200 V; nebulizer gas pressure, 30 psi; desolvation gas flow rate, 13 L/min; desolvation temperature, 350 °C. Both nebulizing and drying gases were supplied by a nitrogen generator. Multiple reaction monitoring (MRM) mode was utilized for quantitative analysis, with specific precursor–product ion transitions detailed in Table 2.

### 2.7. Comprehensive Metabolic Panel

The colorimetric methods, employed on a Mindray BS-420 automated biochemical analyzer (Mindray Bio-Medical Electronics Co., Ltd., Shenzhen, China), were used to measure the serum levels of nine biochemical and physiological parameters in laying hens. These parameters included aspartate transaminase (AST), alanine aminotransferase (ALT), alkaline phosphatase (ALP), globulin (GLB), albumin (ALB), uric acid (UA), glucose (GLU), total cholesterol (TC), and triacylglycerol (TG), adhering to the protocols outlined in the reagent kits provided by BioSino Bio-Technology & Science Inc. (Beijing, China).

### 2.8. Histopathological Examination

Liver, kidney, oviduct, and uterus tissue samples that had been preserved in a fixative solution were embedded in paraffin and then sectioned. After dewaxing, the sections were stained using hematoxylin and eosin (H&E) and subsequently analyzed under a light microscope. In our study, the histopathological examination of liver, kidney, oviduct, and uterus tissues was conducted by two experienced pathologists from the College of Veterinary Medicine, Hunan Agricultural University. The assessment was performed in a blinded manner, meaning that the pathologists were not informed of the group assignments (CN, BP, BPX) of the samples that they were evaluating, so as to prevent any potential bias during the observation and scoring process. Regarding the semi-quantitative scoring criteria, the examination focused on key pathological changes such as inflammatory cell infiltration, steatosis (fat vacuoles), and cellular integrity/architecture. These changes were scored based on their severity and distribution: Inflammatory Cell Infiltration: Scored as 0 (absent), 1 (mild, focal), 2 (moderate, multifocal), or 3 (severe, diffuse). Steatosis (hepatocytes): Scored based on the percentage of affected cells: 0 (<5%), 1 (5–25%), 2 (26–50%), or 3 (>50%). Tubular Epithelial Detachment (Kidney): Scored as 0 (absent), 1 (mild), 2 (moderate), or 3 (severe). General tissue architecture and integrity were assessed qualitatively as either ‘normal’ or ‘altered’.

### 2.9. Data Processing

Residue data from laying hens were recorded and organized using Microsoft Excel. The replicate was consistently treated as the experimental unit throughout the study. All statistical analyses were performed using the mean values of each replicate (n = 6). Data are expressed as replicate means ± standard deviation (SD). A one-way analysis of variance (ANOVA) followed by Duncan’s multiple comparison test was applied, with statistical significance defined at *p* < 0.05.

For the identification of metabolites in laying hens, potential metabolite chromatographic peaks were initially discerned by comparative analysis of MS1 spectra between blank and test samples. Subsequently, a comprehensive investigation of candidate metabolites was conducted through the integrated application of precisely targeted, widely targeted, and untargeted mass spectrometric approaches. Key parameters of all reported ALL and PRO metabolites—including mass-to-charge ratios (*m*/*z*), structural formulae, molecular formulae, and characteristic fragmentation patterns—were systematically compiled through extensive literature curation. Plausible metabolite structures were hypothesized based on established biotransformation pathways, with theoretical *m*/*z* values and molecular formulae calculated using ChemBioDraw Ultra 14.0. Ions demonstrating chromatographic peak intensities ≥ 10^4^ in blank-subtracted samples were prioritized for subsequent MS/MS analysis. To ensure high-confidence identifications, these candidate ions were subjected to fragmentation only when exhibiting a mass error of <5 ppm for the precursor ion. Structural proposals were inferred through comprehensive interpretation of product ion spectra and characteristic fragmentation pathways, leading to putative annotations at MSI Level 2 confidence for all reported metabolites.

## 3. Results

### 3.1. Metabolite Identification

#### 3.1.1. Metabolic Profile of ALL in Laying Hens

##### Fragmentation Pathways of ALL

Under the present chromatographic conditions, ALL exhibited a retention time of 11.3 min and was detected as the protonated ion [M + H]^+^ at *m*/*z* 370.1634 (molecular formula C_21_H_24_NO_5_^+^). As a protopine-type alkaloid, ALL undergoes characteristic fragmentation dominated by retro-Diels–Alder (RDA) cleavage and α-bond fission. The molecular ion fragmented to yield product ions at *m*/*z* 206.0794 and 165.0897. Subsequent losses of OH and CH_4_ from these ions generated fragments at *m*/*z* 189.0772 and 149.0582, respectively. The ion at *m*/*z* 206.0794 further eliminated H_2_O to form *m*/*z* 188.0692. Concurrently, the precursor ion underwent rearrangement followed by sequential neutral losses: dehydration produced *m*/*z* 352.1527, demethylation yielded *m*/*z* 339.1201, and further dehydration generated *m*/*z* 321.1249. The proposed fragmentation pathways are illustrated in Figure 1 (MS1/MS2 spectra) and Figure 2 (fragmentation scheme). Due to the unavailability of authentic standards for the identified metabolites, structural elucidation was based on high-resolution mass spectrometry data and characteristic fragmentation patterns.

##### Tentative Identification of ALL Metabolites in Laying Hens

Subsequent comparative analysis of dosed versus blank control samples using UPLC-Q-TOF-MS tentatively identified nine ALL metabolites in laying hen plasma and fecal samples, designated as ALL-M1 to ALL-M9. Representative MS1 and MS2 spectra and proposed metabolic pathways are presented in Figure 3 and Figure 4, respectively, with Table 3 summarizing critical identification parameters, including retention times, molecular formulae, observed *m*/*z* values, and mass accuracy errors.

(1)Plasma

In addition to the prototype compound ALL, seven metabolites designated as ALL-M1 through ALL-M7 were detected in pooled plasma samples collected from 5 to 120 min post-administration.

The metabolite ALL-M1 exhibited a retention time of 6.7 min. Detected in positive ion mode as the [M + H]^+^ ion at *m*/*z* 356.1427, it displayed a mass decrease of 14 Da compared to the prototype compound ALL. This mass shift indicates the loss of a methylene group, suggesting that ALL-M1 is likely a demethylated metabolite of ALL. Fragmentation of the precursor ion produced a fragment at *m*/*z* 338.1212, corresponding to dehydration. Further sequential losses of methylamine and methanol from this fragment yielded the ion observed at *m*/*z* 275.0552. A distinct fragment ion at *m*/*z* 206.0741 originated from RDA cleavage of the precursor ion. Dehydration of this RDA-derived fragment generated the ion at *m*/*z* 188.0662, while loss of a hydroxyl group produced the ion at *m*/*z* 189.0785. The specific RDA fragmentation pattern and the resulting fragment ions localized the hydroxyl and methoxy substituents to the C9 and C10 positions. Consequently, the structural evidence identifies ALL-M1 as a demethylated derivative of ALL at either the C9 or C10 position.

The metabolite ALL-M2 exhibited a retention time of 4.1 min and an [M + H]^+^ ion at *m*/*z* 532.1784, representing a mass increase of 176 Da compared to ALL-M1. Its major fragment ions at *m*/*z* 356.1403, 338.1277, 188.0666, and 189.0783 were identical to those of ALL-M1. Therefore, ALL-M2 was inferred to be the glucuronide conjugate of ALL-M1.

ALL-M3 displayed a retention time of 9.2 min and an [M + H]^+^ ion at *m*/*z* 372.1765, corresponding to a mass increase of 2 Da relative to the prototype compound ALL. Dehydration of the precursor ion yielded the fragment ion at *m*/*z* 354.1721. The fragment ion at *m*/*z* 208.0862 originated from an RDA cleavage of the parent nucleus; subsequent dehydration of this RDA fragment generated the ion at *m*/*z* 190.0812. The fragment ion at *m*/*z* 149.0607 resulted from the loss of methane from the RDA-derived fragment ion *m*/*z* 165.0897. The fragment ions at *m*/*z* 208.0862 and 149.0607 collectively indicate methylenedioxy ring cleavage at positions C2 and C3 on the ALL scaffold.

ALL-M4 had a retention time of 6.4 min and an [M + H]^+^ ion at *m*/*z* 358.1627, representing a mass decrease of 12 Da relative to ALL and a loss of 14 Da compared to ALL-M3. Dehydration of the precursor ion produced the fragment ion at *m*/*z* 340.1428. The appearance of fragment ions at *m*/*z* 165.0854 and *m*/*z* 194.0762 signifies RDA cleavage of the ALL-M4 structure, with hydroxyl groups located at C2 and C3 and methoxy groups at C9 and C10. The fragment ion at *m*/*z* 176.0685 arose from dehydration of the *m*/*z* 194.0762 ion. Consequently, ALL-M4 was identified as a demethylated derivative of ALL-M3.

ALL-M5 showed a retention time of 5.1 min and an [M + H]^+^ ion at *m*/*z* 534.1958, indicating a mass increase of 176 Da relative to ALL-M4. Its major fragment ions corresponded to those of ALL-M4. Thus, ALL-M5 was inferred to be the glucuronide conjugate of ALL-M4, with the glucuronyl moiety likely substituting a hydroxyl group at either the C2 or C3 position.

ALL-M6 eluted at 9.7 min with an [M + H]^+^ ion at *m*/*z* 386.1543, corresponding to a mass increase of 16 Da compared to ALL. Dehydration of the precursor ion yielded the fragment ion at *m*/*z* 368.1453. This ion subsequently lost a molecule of C_3_H_5_NO to generate the fragment ion at *m*/*z* 297.1032. Concurrently, the ion at *m*/*z* 368.1453 underwent RDA cleavage analogous to ALL, producing the fragment ion at *m*/*z* 222.0790, which underwent further dehydration to yield the ion at *m*/*z* 204.0682. Therefore, ALL-M6 is proposed to be an epoxidized derivative of ALL on ring A or B [19].

ALL-M7 exhibited a retention time of 11.1 min and an [M + H]^+^ ion at *m*/*z* 386.1949, representing a mass increase of 14 Da relative to ALL-M3. Combined with its fragmentation pattern, this suggests ALL-M7 is the methylated derivative of ALL-M3. Fragment ions at *m*/*z* 368.1857, 306.1114, 222.1243, and 204.0965 were generated through dehydration of the precursor ion, loss of methylamine and methanol, RDA cleavage of the precursor ion, and subsequent dehydration of the RDA fragment, respectively.

(2)Fecal Samples

Analysis of fecal samples collected 0–6 h post-administration revealed the presence of seven metabolites. Four of these metabolites—specifically, ALL-M1 through ALL-M4—were also detected in plasma. The remaining three metabolites were designated as ALL-M8 and ALL-M9; their respective primary and secondary mass spectra are presented in Figure 3.

Within the fecal samples, two compounds were detected with identical *m*/*z* values of 372.1727 and 372.1765, exhibiting retention times of 9.2 min and 9.5 min, respectively. One compound corresponded to ALL-M3, while the other represented an isomeric form, designated as ALL-M8. Both isomers exhibited identical fragmentation pathways, with all fragment ions matching precisely. This observation collectively indicates that ALL-M3 and ALL-M8 are positional isomers differing in the locations of one hydroxyl and one methoxy group on the C2 and C3 positions.

Similarly, metabolites at *m*/*z* 356.1427 and 356.1430 were observed with retention times of 6.7 min and 7.9 min, respectively. One metabolite was identified as ALL-M1, and the other as its isomeric counterpart, designated as ALL-M9. The fragmentation patterns of ALL-M1 and ALL-M9 were indistinguishable, with complete agreement in their fragment ion profiles. Consequently, this data suggests that ALL-M9 and ALL-M1 are positional isomers resulting from the alternate positioning of one hydroxyl and one methoxy group on the C9 and C10 positions.

#### 3.1.2. Metabolism of PRO in Laying Hens

##### Fragmentation Pathways of PRO

Under the chromatographic conditions employed, PRO exhibited a retention time of 10.3 min and was detected in its protonated form at *m*/*z* 354.1343, corresponding to the molecular formula C_20_H_20_NO_5_^+^. As a representative protopine alkaloid, PRO underwent characteristic RDA and alpha-cleavage reactions, generating product ions at *m*/*z* 149.0594, 206.0797, and 165.0539. The ion at *m*/*z* 206.0797 subsequently lost a hydroxyl group and a water molecule, yielding fragment ions at *m*/*z* 189.0773 and 188.0696, respectively. Further fragmentation involved the loss of methylamine from PRO, producing the fragment ion at *m*/*z* 323.0933. Concurrently, the presence of a carbonyl group facilitated an intramolecular rearrangement yielding a hydroxyl intermediate, culminating in the loss of water to form the fragment ion at *m*/*z* 336.1242, corresponding to a mass loss of 18.0106 Da from the precursor ion. Figure 5 and Figure 6 depict the primary and secondary mass spectra of PRO, respectively, alongside the proposed fragmentation scheme.

##### Identification of PRO Metabolites in Laying Hens

Employing UPLC-Q-TOF-MS technology, comparative analysis of dosed samples against blank control samples enabled the inference of twelve PRO metabolites in plasma and fecal matrices of laying hens. These metabolites were designated PRO-M1 through PRO-M12. Primary and secondary mass spectra for these metabolites are presented in Figure 7 and Figure 8, respectively, while Table 4 comprehensively summarizes detailed metabolite characteristics, including retention times, molecular formulae, *m*/*z* values, and mass errors.

(1)Plasma

Analysis of pooled plasma samples collected 5 to 120 min post-administration detected the PRO prototype alongside two metabolites, designated PRO-M1 and PRO-M2.

The metabolite PRO-M1 exhibited a retention time of 5.6 min and an *m*/*z* value of 342.1313, which is 12 Da lower than that of the protonated molecular ion of the parent compound PRO. The parent nucleus structure initially underwent dehydration (–18 Da) at a specified collision energy to form the product ion at *m*/*z* 324.1263. Concurrently, PRO-M1 underwent RDA cleavage, yielding product ions at *m*/*z* 194.0732 and *m*/*z* 149.1772. The ion at *m*/*z* 194.0732 subsequently lost H_2_O and OH, forming fragment ions at *m*/*z* 176.0626 and 177.0768, respectively. Analysis of the two RDA fragments suggests that the methylenedioxy bridge at C2 and C3 of PRO-M1 cleaved, resulting in the loss of a CH_2_ molecule and the formation of two neighboring dihydroxy groups. If, following ring cleavage and demethylation, the parent nucleus structure underwent reductive cleavage of an oxygen double bond, the resulting fragments and cleavage patterns were identical. Therefore, PRO-M1 may undergo either of these two reactions, leading to its identification as either the ring-opened demethylated product at C2 and C3 of PRO or the ring-opened, demethylated, and reduced product at C2 and C3 of PRO.

PRO-M2 showed a retention time of 11.1 min and an *m*/*z* value of 386.1941. Based on the fragment ions at *m*/*z* 222.1093 and *m*/*z* 165.0916, it was inferred that this metabolite underwent ring cleavage and methylation at both the C2,3 and C9,10 positions. During mass spectrometric fragmentation, additional ions were formed, including [M–H_2_O + H]^+^ at *m*/*z* 368.1971 and [M–NH_2_CH_3_ + OCH_3_]^+^ at *m*/*z* 306.1283. The ions at *m*/*z* 204.0964 and 205.1066 originated from the product ion at *m*/*z* 222.1093 through dehydration and dehydroxylation, respectively. Based on the MS/MS fragment information, PRO-M2 was identified as the tetramethylated product of PRO at the C2,3,9,10 positions.

(2)Feces

Eleven metabolites were detected in fecal samples collected 0–6 h post-dosing. One metabolite, PRO-M1, had been previously identified in plasma. The remaining ten metabolites were designated PRO-M3 through PRO-M12.

PRO-M1, PRO-M3, and PRO-M4 exhibited *m*/*z* values of 342.1342, 342.1347, and 342.1365, with retention times of 5.6 min, 6.1 min, and 6.4 min, respectively, suggesting that they may be isomers. The fragment ion of PRO-M3 at *m*/*z* 324.1212, 18 Da lower than its precursor ion, indicates a potential dehydration reaction. Characteristic RDA fragment ions at *m*/*z* 206.0793, 188.0703, and 189.0786 suggest ring-opening demethylation of the methylenedioxy bridge at the C9,10 positions. PRO-M3 is therefore proposed as the ring-opened demethylated product of PRO at C9,10. PRO-M4 shared identical major fragment ions and mass spectrometric fragmentation patterns with PRO-M1, indicating that it is either the ring-opened demethylated product or the ring-opened, demethylated, reduced product of PRO at C2,3.

PRO-M5, PRO-M6, and PRO-M7 displayed *m*/*z* values of 518.1633, 518.1626, and 518.1606, with retention times of 3.3 min, 4.3 min, and 4.6 min, respectively. The 176 Da mass increase relative to *m*/*z* 342.1336 suggests the addition of a glucosyl group. As PRO-M7 produced fragment ions identical to those of PRO-M3, it was identified as the glucuronide conjugate of PRO-M3. PRO-M5 and PRO-M6 yielded secondary fragments fully consistent with PRO-M1 and PRO-M4, identifying them as glucuronide conjugates of either PRO-M1 or PRO-M4.

PRO-M8 (*m*/*z* 372.1389, retention time 4.2 min) exhibited an 18 Da increase compared to PRO, tentatively suggesting ring-opening (+2 Da) and hydroxylation (+16 Da). Fragment ions at *m*/*z* 206.1203 and its dehydroxylated product at *m*/*z* 189.0965 indicate that methylenedioxy bridge cleavage did not occur at C9,10 and hydroxylation did not take place on this aromatic ring. The product ion at *m*/*z* 151.0659 likely originated from the RDA fragment *m*/*z* 167.0703 via cyclization (+2 Da) and dehydration (-18 Da), confirming methylenedioxy bridge cleavage at C2,3. The hydroxyl group was positioned at either C11 or C12. Based on this evidence, PRO-M8 is proposed as a ring-opened hydroxylated metabolite of PRO.

PRO-M9 and PRO-M10 (*m*/*z* 532.1796 and 532.1816, retention times 4.3 min and 5.4 min, respectively) were isomers but exhibited distinct fragment ions, indicating different reaction sites. Shared fragment ions at *m*/*z* 356.1482 and *m*/*z* 338.1295 revealed glucosylation (+176 Da) of *m*/*z* 356.1482 and subsequent dehydration in both. Additional fragments of PRO-M9 included *m*/*z* 206.0812, 188.0654, and 189.0785. The RDA fragmentation patterns suggested that the methylenedioxy bridge at C9,10 remained intact. The fragment at *m*/*z* 149.0533 was likely derived from dehydrogenative cyclization of the RDA fragment *m*/*z* 151.0754, indicating substitution at C2,3 by a methoxy group and a hydroxyl group. Therefore, PRO-M9 is proposed as the glucuronide conjugate at either C2 or C3 following ring-opening of PRO. Fragments of PRO-M10 (*m*/*z* 208.0968, 190.0862, 191.0941) suggested modification at the C9,10 methylenedioxy bridge, identifying it as the glucuronide conjugate at either C9 or C10 following ring-opening of PRO.

PRO-M11 (*m*/*z* 534.1925, retention time 5.6 min) yielded a major fragment at *m*/*z* 358.1566. This suggests ring-opening at both C2,3 and C9,10 of PRO, forming two hydroxyl and two methoxy groups, with glucosylation replacing a hydrogen on one hydroxyl group at an undetermined position.

PRO-M12 (*m*/*z* 356.1556, retention time 7.2 min) exhibited secondary fragments similar to those of PRO-M8 and PRO-M9, with an 18 Da mass increase. It underwent RDA cleavage, yielding a product ion at *m*/*z* 238.1479, followed by sequential loss of water to *m*/*z* 220.1079. Fragment *m*/*z* 202.0968 further lost CH2 to form *m*/*z* 188.0983. This fragmentation pattern suggests that, after ring-opening demethylation of the C2,3 methylenedioxy bridge, PRO-M12 underwent dehydration, forming an oxo bridge. The C9,10 methylenedioxy bridge underwent ring-opening, forming a hydroxyl and a methoxy group, concurrent with hydroxylation at C4 and subsequent methylation to form a methoxy group.

### 3.2. Residues of Bopu Powder in Laying Hens’ Tissues

As shown in Table 5, after feeding laying hens a diet supplemented with Bopu Powder for 56 days in the late laying period, no residues of PRO or ALL were detected in the BP group within egg, breast muscle, thigh muscle, skin with fat, abdominal fat, gizzard, plasma, jejunum, ileum, ovary, oviduct, or uterus tissues, with levels below the limit of detection (LOD) of 1 ng/g wet weight. Similarly, in the BPX group, no detectable residues of PRO or ALL were found in breast muscle, thigh muscle, skin with fat, abdominal fat, gizzard, plasma, jejunum, ileum, ovary, oviduct, or uterus tissues. However, residues of PRO at 26.86 ng/g and ALL at 12.29 ng/g were detected in eggs from the BPX group. Low levels of residues were detected in liver and kidney tissues. Specifically, PRO residues were measured at 15.52 ng/g in liver tissue from the BP group, while the BPX group showed PRO residues at 269.49 ng/g and ALL residues at 56.14 ng/g in liver tissue. In kidney tissue, residues of PRO and ALL were 11.21 ng/g and 6.59 ng/g in the BP group, respectively, and 23.62 ng/g and 7.92 ng/g in the BPX group, respectively.

### 3.3. Serum Biochemistry and Histopathological Examination

As shown in Table 6, serum ALT and AST levels in laying hens were significantly decreased in both the BP and BPX groups compared to the CN group. Dietary supplementation with 50 mg/kg Bopu Powder significantly reduced serum TG and TC levels.

Histological analysis of liver sections stained with HE (Figure 9) showed mild inflammatory cell infiltration and a few fat vacuoles present in the CN group. Conversely, the BP and BPX groups demonstrated normal liver architecture, characterized by densely packed hepatocytes and an absence of noticeable lesions. The evaluation of HE-stained kidney sections (Figure 9) revealed an intact renal structure, healthy cells, and distinct nucleoli across all groups (CN, BP, BPX), without any significant lesions. Nonetheless, the CN group exhibited slight detachment of tubular epithelial cells alongside mild inflammatory cell infiltration, while these alterations were less evident in the BP and BPX groups. The analysis of oviduct (Figure 9) and uterine (Figure 9) sections stained with HE showed typical tissue structure, cellular integrity, and closely arranged epithelial layers, with no visible pathological alterations such as inflammatory infiltration in any of the groups. No noteworthy differences were identified among the three experimental groups regarding these reproductive tissues.

## 4. Discussion

### 4.1. Numbers and Types of Metabolites

Comparative analysis of metabolite profiles revealed interspecies differences between laying hens and rats. Following oral administration of ALL to rats, Huang et al. identified 25 metabolites, including isomers. While Phase I metabolic reactions were nearly identical to those observed in our study, Phase II metabolism in rats yielded glucuronidated, glucosylated, and sulfated conjugates. In contrast, the latter two conjugates were undetectable in laying hens [19]. In vitro metabolic data corroborate this pattern: Huang et al. detected eight and five metabolites for ALL and PRO, respectively, in rat liver S9 fractions, compared to only four and three metabolites in porcine liver S9 fractions, respectively, with substantially lower metabolite abundance in the latter system [13]. Wu et al. further characterized ALL metabolism in broiler hepatic microsomes using stable isotope tracing, identifying five key metabolites, with demethylated derivatives predominating [20].

Two factors may explain these discrepancies: First, the relatively low dosage administered in our study resulted in limited abundance of potential metabolites. Many candidates exhibited insufficient signal intensity in primary mass spectrometry for confirmatory secondary mass spectrometry analysis, reducing the number of identifiable metabolites. Minor mass-to-charge ratio deviations occurred in the few detectable metabolites due to low ion abundance. Second, consistent evidence from in vivo and in vitro studies indicates substantial interspecies variation in both metabolite diversity and quantity.

### 4.2. Metabolic Pathways

Structural elucidation of secondary metabolite fragments revealed that both PRO and ALL metabolites share an identical parent nucleus skeleton. These compounds exclusively underwent RDA fragmentation characteristic of protopine alkaloids, followed by intramolecular rearrangement with water elimination via carbonyl groups. Integrating published data on ALL and PRO metabolism with our experimental results [19], the primary metabolic pathways include methylation, demethylation, ring cleavage, oxidation, hydroxylation, and glucuronidation. Notably, these biotransformations predominantly occur at the methylenedioxy or methoxy groups positioned at C2/C3 and C9/C10 of the parent nucleus, and on their associated benzene rings.

### 4.3. Residues of Bopu Powder in Laying Hens’ Tissues

In this study, laying hens were fed diets supplemented with Bopu total alkaloids powder at 50 mg/kg or 500 mg/kg for 56 days. The 50 mg/kg dose was selected based on its established efficacy in improving egg quality and antioxidant capacity without adverse effects [8]. The 500 mg/kg dose was included not only to investigate residue deposition patterns under exaggerated exposure conditions but also to comprehensively evaluate tolerance. The results demonstrated a favorable safety profile for Bopu Powder. No residues of PRO or ALL were detected in edible tissues (eggs, breast muscle, or thigh muscle) at the 50 mg/kg dose. Residues were confined to metabolizing organs (liver and kidneys), consistent with distribution patterns observed in previous studies [14]. At 500 mg/kg, low but detectable residue levels were found in eggs (PRO: 26.86 ng/g; ALL: 12.29 ng/g), indicating potential accumulation at excessive doses, and highlighting the need for a withdrawal period when dosages exceed recommended levels.

These findings confirm a 10-fold safety margin for the 50 mg/kg dose and provide practical guidance for the safe use of Bopu Powder under field conditions. This study supports its application as a long-term feed additive at recommended concentrations and emphasizes adherence to proper dosage regimens to minimize potential risks.

### 4.4. Tolerance Evaluation of Bopu Powder in Laying Hens

During their late production phase, laying hens frequently develop hepatic lipid metabolism disorders and fatty liver syndrome due to intensive egg-laying activity and physiological aging [21]. The serum biochemical indices observed in this study, including reduced ALT and AST levels and improved TG and TC profiles, suggest a potential positive effect of dietary supplementation with 50 mg/kg Bopu Powder on hepatic lipid metabolism and liver function. These improvements may be attributed to the known antioxidant and anti-inflammatory properties of protopine alkaloids, which could help mitigate oxidative stress and modulate lipid metabolism pathways in the liver. Notably, all experimental groups received the same basal diet with identical premix supplementation, ensuring that micronutrients such as selenium and copper did not confound the assessment of hepatic safety.

Histopathological examination of liver and kidney sections revealed no treatment-induced tissue damage at either 50 or 500 mg/kg supplementation levels. Moreover, the supplementation reduced hepatic inflammatory infiltration and steatosis while decreasing renal tubular epithelial cell detachment, indicating potential protective effects on hepatic and renal tissues. These observations align with documented hepatorenal-protective properties of active constituents including PRO and ALL [22,23]. Previous studies established that PRO administration at 11–22 mg/kg body weight effectively counteracts carbon tetrachloride, acetaminophen, and simvastatin-induced hepatotoxicity [24]. In complementary research, Liu et al. demonstrated in broilers that 40 mg/kg dietary supplementation improves hepatic health through inhibition of the TLR4/MyD88/NF-κB/NLRP3 signaling pathway, consequently reducing caspase-1-mediated hepatic pyroptosis [7].

Laying hens’ production involves continuous ovulation and egg-laying activity, where reproductive system health and functionality are critical for maintaining high production performance. In this study, dietary supplementation with 50 or 500 mg/kg Bopu Powder induced no significant pathological alterations in ovarian, oviductal, or uterine tissues based on histological examination, indicating the absence of toxic effects on reproductive organs. Supporting these findings, Yang et al. demonstrated that administering 400 mg/kg (clinical therapeutic dose) or 2000 mg/kg (5-fold clinical dose) to breeder hens for 10 consecutive days negatively impacted neither reproductive performance nor egg fertility and hatchability rates [25]. Furthermore, long-term supplementation at 500 mg/kg (10-fold recommended dose) caused no significant adverse effects on serum biochemical parameters or tissue integrity in laying hens, confirming excellent tolerance at this dosage level.

## 5. Conclusions

This study employed LC-QTOF-MS to identify, for the first time, nine ALL and twelve PRO metabolites in laying hens’ plasma and feces. Among these, two novel metabolites—PRO-M1 (4) and PRO-M12—were characterized, which have not been previously reported in poultry species based on an exhaustive review of the existing literature. Dietary supplementation with 50 mg/kg or 500 mg/kg Bopu Powder demonstrated favorable safety profiles, exhibiting no significant adverse effects on serum biochemical parameters or major organs in late-phase laying hens. No residues were detected in eggs or muscle tissues at the 50 mg/kg dose, demonstrating its safety for long-term use. However, low but detectable residue levels were observed in eggs at the 500 mg/kg dose, suggesting a potential risk of accumulation under exaggerated exposure conditions. These findings highlight the need for further studies to evaluate residue depletion kinetics and establish an appropriate withdrawal period when high-dose supplementation is used.

## Figures and Tables

**Figure 1 vetsci-12-00848-f001:**
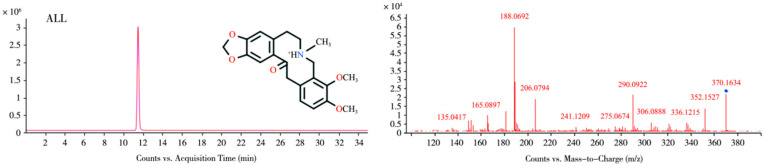
MS and MS/MS spectra of ALL.

**Figure 2 vetsci-12-00848-f002:**
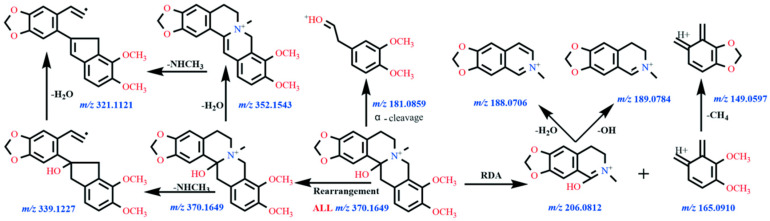
The proposed fragmentation pathways of ALL.

**Figure 3 vetsci-12-00848-f003:**
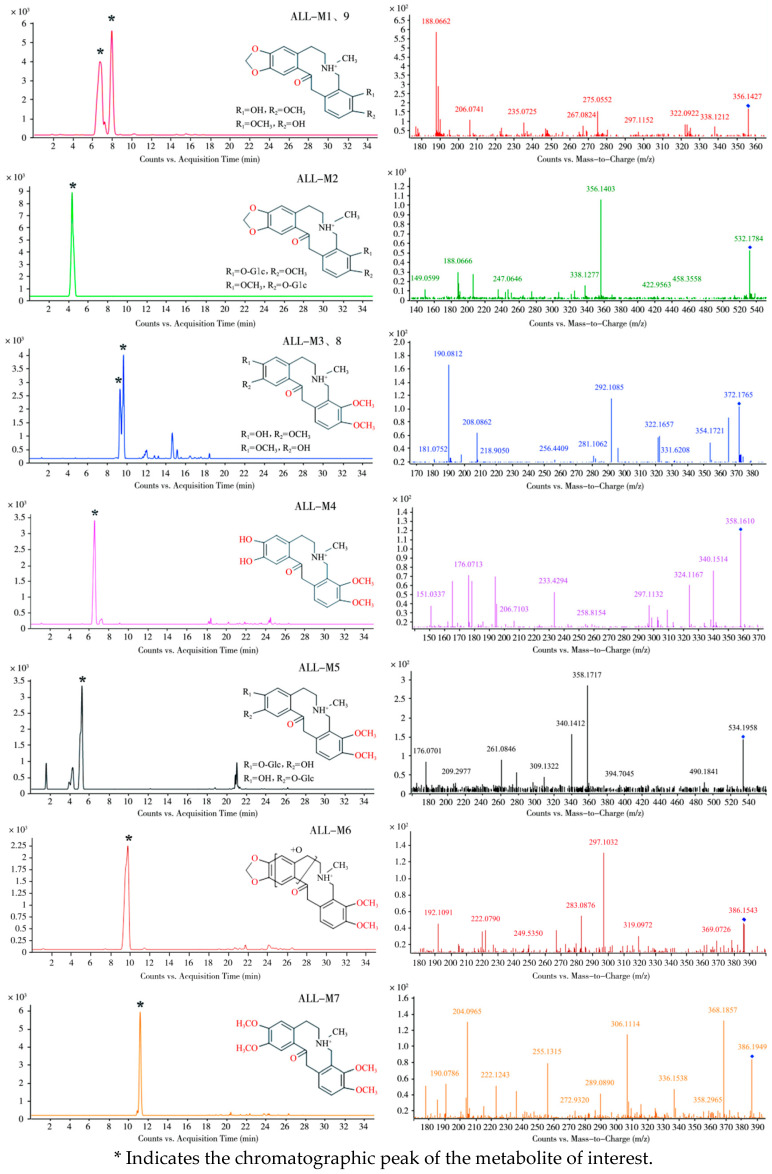
MS and MS/MS spectra for metabolites of ALL.

**Figure 4 vetsci-12-00848-f004:**
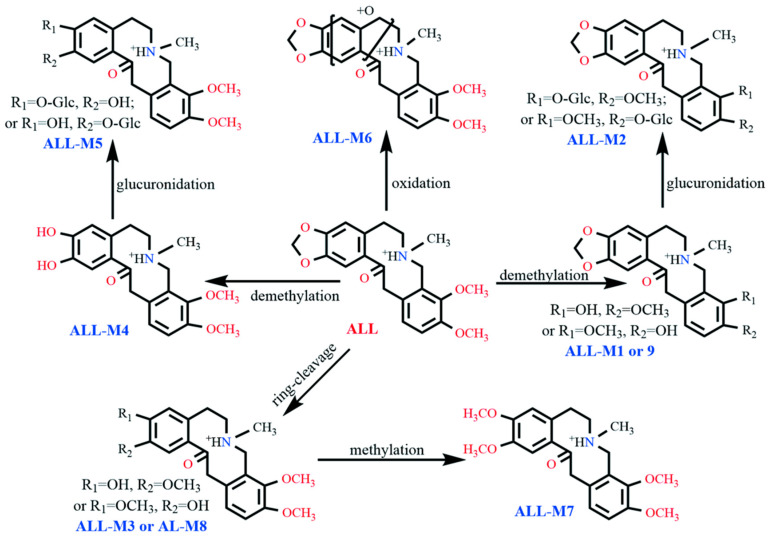
The proposed biotransformation patterns of ALL in vivo.

**Figure 5 vetsci-12-00848-f005:**
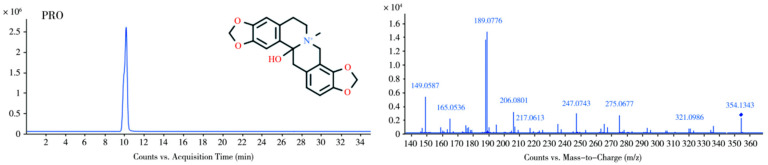
MS and MS/MS spectra of PRO.

**Figure 6 vetsci-12-00848-f006:**
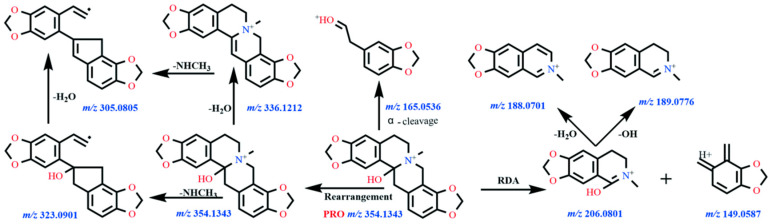
The proposed fragmentation pathways of PRO.

**Figure 7 vetsci-12-00848-f007:**
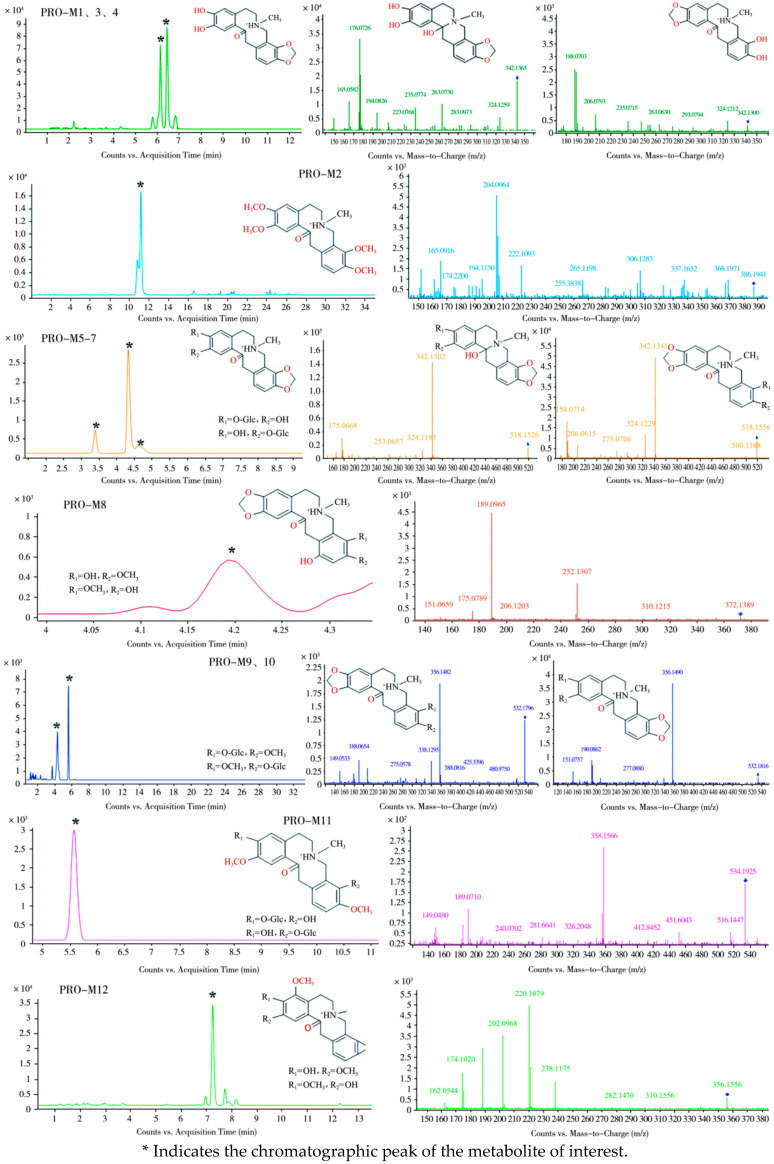
MS and MS/MS spectra for metabolites of PRO.

**Figure 8 vetsci-12-00848-f008:**
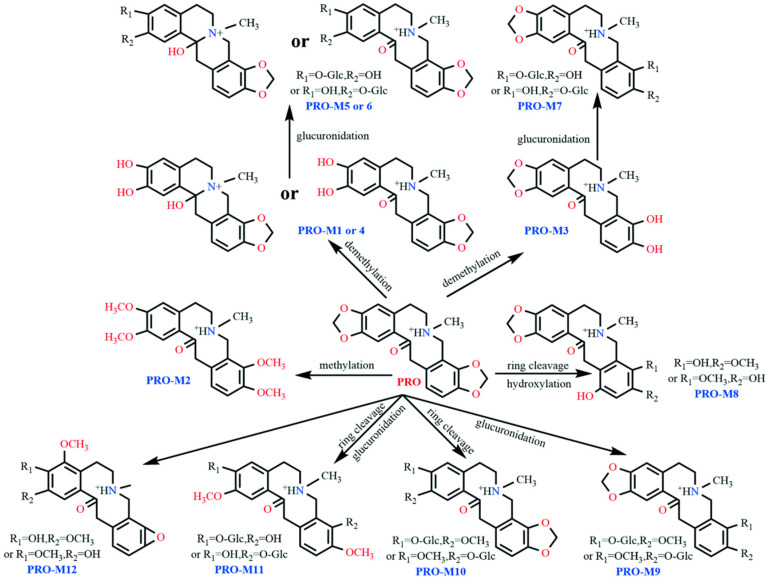
The proposed biotransformation patterns of PRO in vivo.

**Figure 9 vetsci-12-00848-f009:**
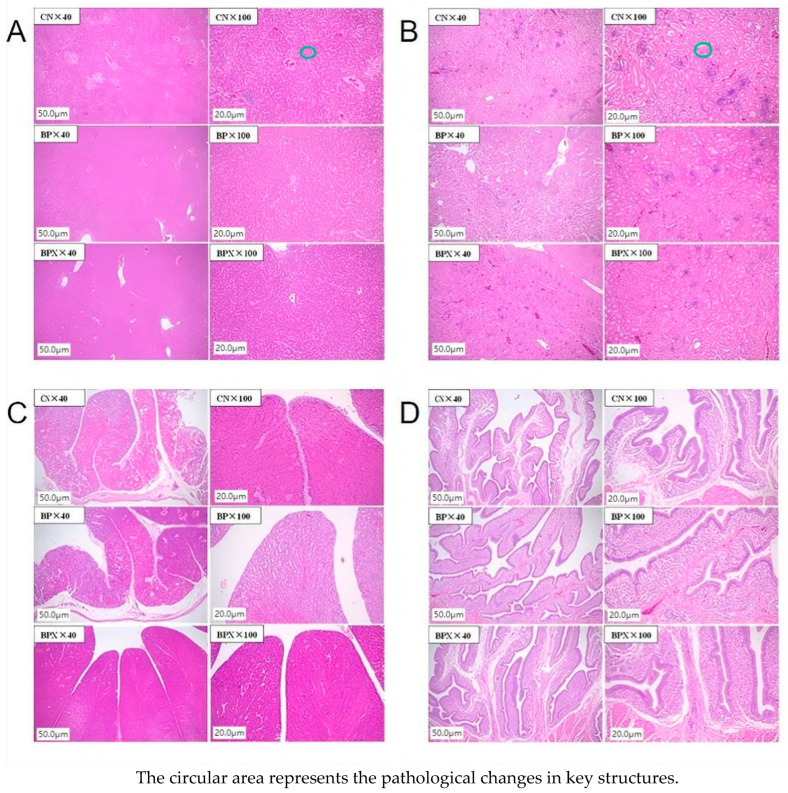
Layer tissue HE sectioning: (**A**) liver; (**B**) kidney; (**C**) oviduct; (**D**) uterus.

**Table 1 vetsci-12-00848-t001:** Composition and nutrient levels of basal diets (air-dry basis, %).

Items	Diets
Ingredients, %	
Corn	58
Soybean meal	27
Soybean oil	2.2
Wheat bran	1.5
Limestone	8.28
CaHPO_4_·2H_2_O	1.5
NaCl	0.2
Na_2_SO_4_	0.2
Met	0.12
50%	0.1
Premix ^1)^	1
Total	100.00
Nutrient Contents ^2)^	
Metabolic energy, MJ/kg	11.37
Crude protein, %	16.86
Crude fiber, %	3.03
Ca, %	3.63
Total phosphorus, %	0.61
Available phosphorus, %	0.35
Lys, %	0.87
Met, %	0.38
Met + Cys, %	0.66

^1)^ The premix provided the following (per kilogram of complete diet) micronutrients: VA 6 000 IU, VD_3_ 2 500 IU, VE 25 mg, VK_3_ 2.25 mg, VB_1_ 1.8 mg, VB_2_ 7 mg, VB_6_ 4 mg, VB_12_ 0.2 mg, *D*-pantothenic acid 12 mg, nicotinic acid 35 mg, biotin 0.14 mg, folic acid 0.8 mg, Cu (as copper sulfate) 11 mg, Zn (as zinc sulfate) 70 mg, Fe (as ferrous sulfate) 60 mg, Mn (as manganese sulfate) 115 mg, Se (as sodium selenite) 0.30 mg, I (as potassium iodide) 0.4 mg. ^2)^ Nutrient levels are calculated values.

**Table 2 vetsci-12-00848-t002:** Mass spectrometry parameters and retention times for TET, PRO, and ALL.

Analyte	Retention Time (min)	Monitored Ion Pair (*m*/*z*)	Fragmentor (V)	Collision Energy (eV)
TET	5.5	356.0→192 * 356.0→165.0	142	26
PRO	3.8	354.1→189.0 * 354.1→149	128	34
ALL	5.0	370.1→188.1 * 370.1→290	114	22

Note: * Quantify ions.

**Table 3 vetsci-12-00848-t003:** Metabolite identification results of ALL in vivo.

Metabolite	Retention Time	Proposed Formula	Measured	Calculated	Error (ppm)	Fragment Ions	Fragmentation Pathway
ALL	11.3	C_21_H_24_NO_5_^+^	370.1634	370.1649	−4.05	352.1527, 206.079, 165.0897, 188.0692	Absent
ALL-M1	6.7	C_20_H_22_NO_5_^+^	356.1427	356.1492	−18.25	338.1212, 275.055, 188.0662, 189.0785	Demethylation
ALL-M2	4.1	C_26_H_30_NO_11_^+^	532.1784	532.1813	−5.45	356.1403, 338.1277, 188.0666, 189.0783	Glucuronidation
ALL-M3	9.2	C_21_H_26_NO_5_^+^	372.1765	372.1805	−10.75	354.1721, 208.0862, 208.0862, 149.0607	Ring Cleavage
ALL-M4	6.4	C_20_H_24_NO_5_^+^	358.1627	358.1649	−6.14	340.1428, 194.0762, 165.0854, 176.0685	Ring Cleavage and Demethylation
ALL-M5	5.1	C_26_H_32_NO_11_^+^	534.1958	534.1970	−2.25	358.1717, 340.1412, 176.0701	Glucuronidation
ALL-M6	9.7	C_21_H_24_NO_6_^+^	386.1543	386.1593	−12.95	368.1453, 297.1032, 222.0790, 204.0682	Oxidation
ALL-M7	11.1	C_22_H2_8_NO_5_^+^	386.1949	386.1962	−3.37	368.1857, 306.1114, 222.1243, 204.0965	Methylation
ALL-M8	9.5	C_21_H_26_NO_5_^+^	372.1727	372.1805	−20.96	354.1720, 208.0797, 189.0758, 149.0607	Ring Cleavage
ALL-M9	7.9	C_20_H_22_NO_5_^+^	356.1445	356.1492	−13.20	338.1220, 275.0637, 188.0701, 189.0780	Demethylation

**Table 4 vetsci-12-00848-t004:** Metabolite identification results of PRO in vivo.

Metabolite	Retention Time	Proposed Formula	Measured	Calculated	Error (ppm)	Fragment Ions	Fragmentation Pathway
PRO	10.3	C_20_H_20_NO_5_^+^	354.1343	354.1336	1.98	336.124, 149.0594, 206.0797 165.053, 189.0773, 188.0696	Absent
PRO-M1	5.6	C_19_H_20_NO_5_^+^	342.1342	342.1336	1.75	324.126, 194.0732, 149.1772, 176.0626, 177.0768	Ring cleavage, demethylation, or rearrangement; ring cleavage and demethylation
PRO-M2	11.1	C_22_H_28_NO_5_^+^	386.1941	386.1962	−5.44	368.197, 222.1093, 165.0916, 204.0964, 205.1066	Methylation
PRO-M3	6.1	C_19_H_20_NO_5_^+^	342.1300	342.1336	−10.52	324.121, 206.0793, 188.0703, 189.0786	Ring cleavage and demethylation
PRO-M4	6.4	C_19_H_20_NO_5_^+^	342.1365	342.1336	8.48	324.123, 194.0730, 149.1692, 176.061, 177.0765	Ring cleavage, demethylation, or rearrangement; ring cleavage and demethylation
PRO-M5	3.3	C_25_H_28_NO_11_^+^	518.1633	518.1657	−4.63	500.155, 342.1335, 324.1236, 176.0701	Glucuronidation
PRO-M6	4.3	C_25_H_28_NO_11_^+^	518.1626	518.1657	−5.98	342.130, 324.1193, 194.0785, 176.0668	Glucuronidation
PRO-M7	4.6	C_25_H_28_NO_11_^+^	518.1656	518.1657	−0.19	342.134, 324.1229, 206.0816, 188.0714	Glucuronidation
PRO-M8	4.2	C_20_H_22_NO_6_^+^	372.1389	372.1442	−14.24	206.120, 189.0965, 151.0659, 167.0703	Ring cleavage and hydroxylation
PRO-M9	4.3	C_26_H_30_NO_11_^+^	532.1796	532.1813	−3.19	356.148, 338.1295, 206.0812, 188.0654, 189.0785	Glucuronidation
PRO-M10	5.4	C_26_H_30_NO_11_^+^	532.1816	532.1813	0.56	356.148, 338.1295, 208.0968, 190.0862, 191.0941	Glucuronidation
PRO-M11	5.6	C_26_H_32_NO_11_^+^	534.1925	534.1970	−8.42	358.1566	Glucuronidation
PRO-M12	7.2	C_20_H_22_NO_5_^+^	356.1556	356.1492	17.97	238.1479, 220.1079, 202.0968, 188.0983	Methylation, etc.

**Table 5 vetsci-12-00848-t005:** The residues of PRO and ALL in the tissues of laying hens (ng/g, *n* = 6, mean ± sd).

Tissues	Analytes	CN	BP	BPX
Eggs	PRO	ND	ND	26.86 ± 15.12
	ALL	ND	ND	12.29 ± 4.69
Breast muscle	PRO	ND	ND	ND
	ALL	ND	ND	ND
Thigh muscle	PRO	ND	ND	ND
	ALL	ND	ND	ND
Skin with fat	PRO	ND	ND	ND
	ALL	ND	ND	ND
Abdominal fat	PRO	ND	ND	ND
	ALL	ND	ND	ND
Gizzard	PRO	ND	ND	ND
	ALL	ND	ND	ND
Plasma	PRO	ND	ND	ND
	ALL	ND	ND	ND
Liver	PRO	ND	ND	ND
	ALL	ND	ND	56.14 ± 9.39
Kidney	PRO	ND	11.21 ± 8.87	23.62 ± 6.94
	ALL	ND	6.59 ± 4.61	7.92 ± 4.12
Jejunum	PRO	ND	ND	ND
	ALL	ND	ND	ND
Ileum	PRO	ND	ND	ND
	ALL	ND	ND	ND
Ovary	PRO	ND	ND	ND
	ALL	ND	ND	ND
Oviduct	PRO	ND	ND	ND
	ALL	ND	ND	ND
Uterus	PRO	ND	ND	ND
	ALL	ND	ND	ND

ND: not detected.

**Table 6 vetsci-12-00848-t006:** Effects of Bopu Powder on plasma biochemical parameter indices of laying hens.

Item	CN	BP	BPX	SEM	*p*-Value
ALT, U/L	62.48 ^b^	38.30 ^a^	32.65 ^a^	7.738	0.032
AST, U/L	194.10 ^b^	130.62 ^a^	126.62 ^a^	5.58	0.035
LDH, U/L	347.38	236.50	235.80	17.202	0.161
T-Bil, μmol/L	46.40	47.90	49.26	0.506	0.077
ALB, g/L	5.97	9.22	8.63	0.669	0.192
GLU, mmol/L	9.04	11.69	9.71	0.624	0.320
UREA, mmol/L	1.06	2.06	0.97	0.221	0.205
TG, mmol/L	16.67 ^b^	5.71 ^a^	10.01 ^ab^	1.722	0.039
TC, mmol/L	6.36 ^b^	3.82 ^a^	5.44 ^ab^	0.305	0.008
HDL-C, mmol/L	0.156 ^b^	0.216 ^a^	0.164 ^b^	0.007	<0.001
LDL-C, mmol/L	0.664	0.501	0.434	0.098	0.713

^a,b^ Different superscript letters indicate statistically significant differences at *p* < 0.05.

## Data Availability

The original contributions presented in this study are included in the article. Further inquiries can be directed to the corresponding author.

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
