# Peer review of "Metabolic Profiling, Tissue Distribution, and Tolerance Assessment of Bopu Powder in Laying Hens Following Long-Term Dietary Administration"

_vetsci, 2025, doi:10.3390/vetsci12090848_

Round 1
Reviewer 1 Report
Comments and Suggestions for Authors
Manuscript entitled "Metabolic Profiling, Tissue Distribution and Tolerance Assess- 2
ment of Bopu Powder in Laying Hens Following Long-Term 3
Dietary Administration" is well structured and presented in details related to obtained material and methods and results. However, some minor changes are needed as follows:
Lines 46 to 51: Authors used reference Guo et al. (2013) two times in one sentence which is unnecessary. Please correct this for Huang et al (line 51) and others in Introduction and Discussion sections.
Material and methods are presented in details. Some of the procedures are well known. However, it can be useful for readers.
Lines 516 to 519. It is not necessary to indicate level of significance (P<0.05) since you defined level of significance in data processing section.
Conclusion should emphases the importance of this study as it characterize, for the first time, the metabolic disposition of the primary active constituents PRO and ALL from Bopu powder in laying hens (as it was done in last sentence of Abstract).
Reviewer 2 Report
Comments and Suggestions for Authors
- Residues were detected in eggs from the BPX group (PRO 26.86 ng/g; ALL 12.29 ng/g) but not in the 50 mg/kg BP group. Please clarify whether this pattern reflects a dose–response or dose-linearity effect, and if any threshold concentration for egg deposition is implied.
- Only the liver, kidney, oviduct, and uterus were examined histologically. Please explain the rationale for excluding intestinal tissues from histological analysis, given their potential role in alkaloid metabolism and absorption.
- The CN group liver exhibited mild inflammatory cell infiltration and fatty vacuoles. Please discuss whether this represents background pathology commonly observed in late-phase laying hens, and how comparability between groups was ensured.
- All residue and histopathology sampling was performed at the end of the 56-day period. Please clarify whether temporal dynamics of accumulation and clearance were considered, and why intermediate time points were not included.
- In relation to the 7-day feeding study by Liu et al. (2025), please elaborate on whether the present 56-day data reveal any novel residue distribution patterns or altered tissue accumulation trends.
- Please revise inconsistent terminology throughout the manuscript, such as 'Bopu Powder' vs. 'Bopu powder'."
- Define all abbreviations at first use (e.g., Bopu powder low-dose group [BP], Bopu powder high-dose group [BPX], protopine [PRO], allocryptopine [ALL]) and maintain consistency thereafter.
- Please standardize unit notations, e.g., 'ng/g' vs. 'ng·g⁻¹', 'mg/kg' vs. 'mg·kg⁻¹'.
- Could the detected residue levels in liver and kidney tissues (e.g., 269.49 ng·g⁻¹ PRO in BPX group liver) be compared with existing food safety standards such as Maximum Residue Limits (MRLs)?"
- Please correct grammatical and spelling errors throughout the manuscript.
- For instance, line 286, it is recommended that no abbreviations be used in all tables and figures. Additionally, abbreviations and full names should be included in the table notes and legends.
- Figure 9, it is recommended to mark the key structures (such as renal tubules).
Reviewer 3 Report
Comments and Suggestions for Authors
Journal: Veterinary Sciences
Manuscript ID: vetsci-3828315
Type of manuscript: Article
Title: Metabolic Profiling, Tissue Distribution and Tolerance Assessment of
Bopu Powder in Laying Hens following Long-term Dietary Administration
Authors: Hongting Wang, Xinhao Wang, Jiaxin Xu, Zihui Yang, Zhen Dong,
Jianguo Zeng *, Hua Liu *
The authors aimed to elucidate the metabolic profile and safety of
Bopu powder in laying hens, focusing on its principal alkaloids protopine
(PRO) and allocryptopine (ALL). They illustrated that dietary supplementation with 50 mg/kg or 500 mg/kg Bopu powder demonstrated favorable safety profiles, exhibiting no significant adverse effects on serum biochemical parameters or major organs in late-phase laying hens. Even though the work is interesting, several critical concerns were found in the whole manuscript and require a major revision.
Please refer to the comments in the PDF file and revise accordingly before further consideration.

Round 2
Reviewer 3 Report
Comments and Suggestions for Authors
The authors revised accordingly previous comments and no further comments.